# Dignity in Care of Older Patients with Cancer in Korea: A Hybrid Model Concept Analysis

**DOI:** 10.3390/healthcare13222935

**Published:** 2025-11-16

**Authors:** Yun Sil Ahn, Pok-Ja Oh, Gye Jeong Yeom

**Affiliations:** 1Cancer Center, VHS Medical Center, Seoul 05368, Republic of Korea; 2College of Nursing, Sahmyook University, Seoul 01795, Republic of Korea; ohpj@syu.ac.kr; 3Department of Nursing Science, JEI University, Incheon 22573, Republic of Korea; salt0402@jeiu.ac.kr

**Keywords:** older patients, respect, care, neoplasms, concept analysis, dignity in care, cancer patient

## Abstract

**Highlights:**

**What are the main findings?**
Dignity in care for older cancer patients encompasses multidimensional attributes, including intrinsic, relational, social, illness-related, and professional dimensions.Older patients maintain dignity through free will, proactive coping, and supportive relationships with nurses, family, and peers, while nurses emphasize ethical attitudes, respect, and professional competency.

**What is the implication of the main finding?**
Preserving dignity enhances older patients’ autonomy, self-esteem, and quality of life, even amid illness and dependency.Nurses and healthcare systems play a pivotal role in sustaining dignity by fostering respectful communication, ensuring systemic support, and empowering patient participation in decision-making.

**Abstract:**

**Background/Objectives**: This study explores the concept of dignity in care for older patients with cancer in Korea using the hybrid model. **Methods**: A three-phase hybrid model approach was employed for concept analysis. In the theoretical phase, a literature review was conducted to determine the attributes and definition of dignity in care. In the fieldwork phase, the practicability of the defined concept was assessed through practical observations. In the final analysis phase, findings from the theoretical and fieldwork phases were compared and synthesized to validate the attributes and definition of the concept. **Results**: Four dimensions of dignity in care were found identified: intrinsic, relational, social, and illness-related. Professional dimension was added based on nurses’ perspective. Attributes of dignity in care for older cancer patients include eight key elements: personal identity, a deepened sense of value and meaning of life, respect, relationships (with medical staff, family, and fellow patients), support for society’s care policies, systemic support from healthcare systems, free will and choice, and proactive coping strategies. For nurses, dignity in care involves seven attributes: understanding and respecting human values, ethical and moral attitudes, interaction-based communication through the cultivation of rapport, systemic support from healthcare systems, protection of dignity, activities promoting dignity, and professional competency. **Conclusions**: This study provides concept definitions and attributes for dignity in care, equipping clinical nurses with assessment tools to better understand and enhance the dignity of older cancer patients.

## 1. Introduction

Humans experience intrinsic dignity, rooted in personal values, and extrinsic dignity, shaped by cultural and social relationships. External dignity can be upheld or violated when interacting with others [1]. Dignity encompasses personal values, respect, freedom, and responsibility, and is influenced by culture and society [2]. True dignity is preserved when individuals are empowered to realize their full potential by recognizing their right to self-actualization [3]. As a fundamental and absolute human value, dignity should be preserved and respected for all patients [4], and because it can be easily compromised, maintaining dignity within healthcare settings is of utmost importance.

However, illness often makes individuals dependent on others, limiting their ability to manage their daily lives and restricting their autonomy [5]. In addition to dealing with the physical symptoms of life-threatening cancer, older cancer patients face various psychological challenges, such as fear of recurrence, anxiety, demoralization, despair, and depression [6]. Functional decline further restricts their autonomy, making it increasingly difficult to maintain their dignity [7]. Older cancer patients, compared with younger ones, are more vulnerable to a loss of autonomy caused by chronic illnesses or physical disabilities, resulting in prolonged hospital stays and increased care dependency [8]. Their dignity can be recognized or compromised depending on the environment, social support available, attitudes of caregivers, and information provided [9].

Advancements in medical technology and improvements in living standards [5] have increased the number of older patients with cancer in Korea, thereby intensifying the demand for nursing care in clinical settings. Furthermore, older adults with cancer are at increased risk of social and familial isolation due to the shift toward nuclear families and changing social structures. The loss of relationships through bereavement, combined with declining physical, visual, and auditory functions, reduces their activity levels and underscores the need for dignity in care. Declining physical and sensory functions further limit their activity, underscoring the need for care that respects and preserves their dignity.

Dignity has long been recognized as a fundamental and unique humanistic concept in nursing, grounded in the basic principle of respecting human beings [10]. Safeguarding patients’ dignity is a core element of nursing, prioritizing human value above all other considerations [11]. Nurses play a pivotal role in preserving patients’ dignity by adopting respectful attitudes, building rapport with patients, attentively listening to their stories, addressing their needs, and encouraging their individual activities [12]. Through effective communication, nurses manage symptoms, provide appropriate environmental and social support, and share disease-related information that empowers patients to make independent treatment decisions [9]. Moreover, by preserving human values and demonstrating ethical and moral attitudes, nurses exemplify professionalism in their care, respecting patients’ autonomy and independence while actively maintaining and enhancing their dignity [13]. Dignity in care refers to both patients’ experience of dignity while receiving care and nurses’ provision of care that promotes or enhances patients’ experience of dignity [9,12,13].

While nurses should possess a clear understanding of dignity in care—which is rooted in the multidimensional and philosophical concept of human dignity influenced by personal, social, and cultural factors [14]—urgent efforts are needed to identify the characteristics and essential attributes of dignity in care for this demographic, thereby providing conceptual clarity and enhancing nursing practice.

Driven by a paradigm shift in healthcare environments and the growing demands for maintaining patient dignity worldwide, various studies have analyzed dignity in patient care and its related concepts [10,14,15]. Preserving patient dignity involves respecting their individual autonomy through the provision of nursing care that considers their unique characteristics and cultural diversity [15]. This approach has been shown to enhance the resilience needed to overcome illness by imbuing life with meaning [16]. The concept of patient dignity is recognized as a cornerstone of healthcare services, positively influencing health outcomes by fostering greater patient participation in decision-making, enhancing their self-esteem, and reinforcing their sense of control over their own lives [17]. For individuals with chronic and progressive diseases, dignity contributes to reduced physical and psychological suffering, improved quality of life, the pursuit of purpose and values, and greater self-esteem [18]. Older cancer patients strive to maintain their quality of life and sustain hope for recovery through treatment [19]. By making independent decisions about their care, they find value and meaning in life, strengthening their autonomy and preserving their dignity [20].

These findings suggest that patients perceive their dignity as protected when their individuality and privacy are respected and when they are empowered with authority over their roles. This recognition enables them to proactively pursue the value and meaning of their lives. Moreover, as it is rooted in mutual respect, dignity is mutually recognized by patients and nurses through the caregiving process [10]. Nurses’ ethical competence, education, and awareness play a crucial role in fostering dignity, whereas challenging environments and organizational shortcomings can undermine it [21]. Therefore, nurses must provide care that preserves dignity while considering the physical environment and cultural differences among their patients.

Both domestic and international research on dignity in care has tended to focus on general patients [22], older adults [23], and terminally ill cancer patients [24], with an emphasis on developing conceptual frameworks and assessment tools [22,23]. Meanwhile, few studies have examined the specific attributes and dimensions of dignity in care for older cancer patients at clinical. Additionally, older cancer patients often encounter challenges in their daily lives due to aging and comorbidities, increasing their dependency on care. This highlights the need to explore how the attributes of dignity in care for older cancer patients are shaped by individual values and beliefs, societal and cultural contexts, and dynamic relationships with others.

Addressing this gap, this study investigates the characteristics of dignity in care, which may have a significant impact on treatment outcomes [17] and quality of life [18] for older cancer patients. In doing so, this study provides foundational data to guide nursing activities that enhance patient dignity and contribute to the improvement of overall healthcare services.

The hybrid model [25], which reflects social and cultural characteristics through participants’ experiences and translates highly abstract and ambiguous concepts into practice, is an appropriate concept analysis method for applying the multidimensional, abstract, and ambiguous concept of dignity in care for older cancer patients to clinical practice. Therefore, this study aims to analyze the concept of dignity in care for older patients with cancer in the Korean clinical context using the hybrid model developed by Schwartz-Barcott and Kim [25].

## 2. Materials and Methods

### 2.1. Study Design

To analyze the concept of dignity in care for older patients with cancer in clinical settings, this study employed the hybrid model of concept analysis [25], which consists of three distinct phases: theoretical, fieldwork, and final analysis.

In the theoretical phase, a comprehensive literature review was conducted to identify potential attributes and definitions of dignity in care. In the fieldwork phase, the concept’s practical applicability was assessed through empirical observations and data collection. In the final analysis phase, findings from the theoretical and fieldwork phases were compared and synthesized to refine and confirm the attributes and definitions of dignity in care for older cancer patients.

#### 2.1.1. The Theoretical Phase

Review of literature

The theoretical phase focused on establishing the conceptual definition and identifying the attributes of dignity in care for older cancer patients. Accordingly, a systematic literature search was conducted, and the selected studies were thoroughly reviewed. Subsequent content analysis resulted in the identification of the dimensions, attributes, and a provisional definition of the concept of dignity in care.

The literature search focused on material published between 2000, when research on dignity in care gained momentum with Chochinov’s study [26], and January 2024. For Korean studies, keywords corresponding to “dignity in care,” “dignity nursing,” “dignity,” and “perception of dignity” were used to search Korean databases, including the Korean Studies Information Service System, DBpia, and the Research Information Sharing Service. For international studies, the search string “older patients and elder patients and respect and care and neoplasms and cancer and dignity care and dignity in care” was used to search various databases, including Google Scholar, PubMed, Embase, and the Cochrane Library.

The literature review extended beyond nursing to incorporate related disciplines, such as law and ethics. Questions guiding the systematic review included: What is dignity in care? How is dignity in care defined? How can dignity in care be conceptualized? What are the components of dignity in care? What are the methods for measuring dignity in care? How can the feasibility of measurement be improved?

Based on this systematic review, provisional attributes and a definition of dignity in care were derived. The search identified 663 Korean articles and 7694 foreign articles. In alignment with the aim of exploring the attributes of dignity in care, the main inclusion criteria combined terms for dignity in care and presence of neoplasms [27]. After removing duplicates and excluding irrelevant articles or those unsuitable for exploring conceptual definitions, a total of 16 articles were included in the final review (Figure 1).

#### 2.1.2. The Fieldwork Phase

Data collection

This study sought to establish a theoretical foundation for dignity in care for older cancer patients by identifying its attributes. Recognizing the interactive nature of care, these attributes were first explored from the perspective of older cancer patients and then examined through the lens of the nurses providing their care. Specifically, individual in-depth interviews were conducted with six older cancer patients, with each participant attending one or two interview sessions. These interviews provided detailed insights into the patients’ emotions, perspectives, attitudes, behaviors, and values. Additionally, two focus groups, each comprising five nurses, were formed. One focus group interview (FGI) was conducted with each group to gain a deeper understanding of their opinions and experiences and to harness the advantages of group interaction.

(1)Setting and participants

Individual interviews with patients with cancer and nurses at a general hospital in South Korea were conducted in Korean from 15 February to 28 February 2024, and focus group interviews (FGIs) were conducted on 5 March and 13 March 2024 (Table 1). This study was conducted after obtaining prior informed consent from participants who voluntarily agreed to the study’s purpose and procedures. Using snowball sampling, six older cancer patients and ten nurses who met the inclusion criteria were recruited and enrolled in this study. To this end, the study recruited two participant groups: (1) older patients with cancer (≥65 years) as the primary focus to explore their experiences of dignified care, with inclusion criteria being a confirmed cancer diagnosis and exclusion criteria including cognitive impairment, diagnosed depression or other mental disorders, and inability to communicate in Korean; and (2) nurses who provide care to these patients, with at least one year of clinical experience [28], in order to examine their perspectives and experiences regarding dignified care and to capture the effects of group interactions in the clinical setting. With respect to concept development using the hybrid model, if the concept under investigation has the individual level as its unit of analysis, six participants are deemed sufficient [25].

FGIs are more efficient than individual interviews and provide opportunities for group interactions, enabling participants to share and reflect on their opinions and experiences, thereby offering deeper insights [29]. This study conducted FGIs to explore interactions related to dignified care for older cancer patients from the perspective of nurses. Using snowball sampling, this study selected ten nurses and divided them into two focus groups of five participants each.

(2)In-depth interviews and Focus group interviews

In-depth interviews were scheduled at times convenient for participants. Each interview lasted approximately 60 min and was guided by a semi-structured questionnaire. Each focus group interview lasted 90 and 70 min, respectively. Interviews continued until data saturation was achieved, with recurring themes consistently emerging from participants’ responses. Non-verbal expressions were documented in field notes, while verbal responses were audio-recorded with participants’ consent. The recorded interviews were transcribed immediately after each session to ensure accuracy and completeness.

Key questions addressed during the in-depth interviews included:What does dignity mean to you?Can you describe your experiences with dignity in care?How did dignity in care influence your emotions, thoughts, and behaviors?What do you think dignity in care entails?What aspects of dignity in care were beneficial?What aspects of dignity in care were counterproductive?How was dignity in care delivered to you?What do you think could improve dignity in care?
Data analysis


The collected data were analyzed using the qualitative content analysis method of Elo and Kyngäs [30], which systematically organizes and condenses complex descriptions into meaningful themes and categories within a structured classification framework, with nurses’ and patients’ data analyzed separately. The researchers meticulously reviewed the transcribed data, field notes, audio recordings, and observations multiple times to extract significant statements related to dignity in care for older cancer patients. These statements were categorized, reviewed, and analyzed iteratively. Each researcher conducted the analysis independently, followed by a collaborative comparison and discussion to achieve consensus. To ensure the trustworthiness and rigor of the study, methodological and investigator triangulation were employed. Data were collected through individual in-depth interviews with older patients with cancer and focus group discussions with nurses to obtain multiple perspectives on dignity in care. The research team independently analyzed and discussed the findings to minimize researcher bias and enhance credibility. Data saturation was achieved when no new themes or attributes emerged from the additional interviews, confirming the adequacy of the sample and the comprehensiveness of the data. This systematic process enabled an in-depth examination of the definitions, dimensions, and attributes of dignity in care for older cancer patients. The analysis ultimately led to the identification of subcategories, which were subsequently grouped to classify the broader attributes and dimensions of the concept.

#### 2.1.3. Final Analysis Phase

In the final analysis phase, the findings from the theoretical and fieldwork phases were integrated and consolidated. The operational definitions and criteria of dignity in care for older cancer patients established via the literature review in the theoretical phase were systematically compared and analyzed alongside the definitions and attributes identified in the fieldwork phase. This integrative process resulted in the confirmation and finalization of the meanings, dimensions, and attributes of dignity in care for older cancer patients. Each category was clarified and labeled to reflect the specific circumstances of older cancer patients and the experiences of their caregivers.

Ethical considerations

This study was approved by the Institutional Review Board of the VHS Medical Center of Korea (BOHUN 2023-09-001-002) and conducted in accordance with the Declaration of Helsinki, national research ethics guidelines, and the policies of the Center’s ethics committee. Before data collection, the researchers explained the study’s purpose and procedures to older cancer patients and nurses, obtaining voluntary participation. Participants were informed of their right to withdraw at any time, and anonymity and confidentiality were strictly assured. The purpose and the process of the study were clearly explained to all participants, and informed consent was obtained from them. Participants were free to withdraw from the interview at any stage of the study. They were also assured that their voices would be recorded and typed immediately after the interview and kept confidential.

Rigor

To ensure the validity and reliability of data collection, the researchers leveraged their extensive experience and expertise in dignity in care. The first author, who had previously participated in a research project analyzing the concept of dignity in nursing and attended qualitative research conferences and seminars, made significant contributions to the study. The co-authors brought additional credibility through their teaching experience in dignity-related nursing and participation in qualitative research seminars, enhancing their proficiency in qualitative data analysis. Adhering to qualitative research rigor standards, the team ensured integrity, consistency, and reflexivity throughout the analysis. They employed a systematic, step-by-step approach and engaged in ongoing discussions to establish mutually agreed-upon concepts, further enhancing the study’s credibility. A nursing professor with expertise in similar studies and a specialist oncology nurse with over 20 years of experience were consulted to verify the alignment and accuracy of the dimensions and attributes identified in this study.

## 3. Results

### 3.1. Theoretical Phase

#### 3.1.1. The Definition of Dignity in Care

The Standard Korean Language Dictionary defines “dignity” as “an inalienably high and solemn state of being,” and “care” as “an act of maintaining or promoting healthy living, regardless of health status, or aiding in the recovery of health” [31]. The Oxford English Dictionary (OED) [32] provides four definitions of “dignity,” the first and fourth of which are most relevant to the context of this study: “(1) The quality of being worthy or honorable; worthiness, worth, nobleness, excellence” and “(4) Nobility or befitting elevation of aspect, manner, or style; becoming or fit stateliness, gravity.” Among the 12 definitions of “care” in the OED, the following align with this study: “An object or matter of care, concern, or solicitude (1590–)” and “The attention and treatment given to a patient by a doctor or other health worker (1968–).” Synthesizing these dictionary definitions, “dignity in care” can be interpreted as “acts of protecting and assisting individuals to maintain healthy living, grounded in the inherent value and respect for human beings.”

#### 3.1.2. Use of the Concept in Other Disciplines

Given the intrinsic and essential value of humans as individuals encapsulated by the concept of “dignity,” many countries have enshrined this principle in their constitutions [33]. Dignity is a central theme in various disciplines, such as theology, philosophy, law, and political science, addressing the principles of equality and justice through universal respect for all humans. In 1994, the World Health Organization [34] declared that all individuals have the right to be respected as human beings and to die with dignity. Chochinov [26] highlighted the significance of preserving dignity in the care of terminally ill patients within the medical context. He argued that maintaining dignity promotes symptom relief, psychological well-being, and spiritual peace. Chochinov [26] expanded the concept of dignity in care to include (1) illness-related dignity (reducing suffering and maintaining functional independence), (2) intrinsic dignity (promoting autonomy and recognition of personal dignity), and (3) social dignity (protecting privacy and fostering respectful interpersonal interactions). Across disciplines, dignity in care can be understood as actions that safeguard and uphold human dignity through caregiving, particularly for individuals experiencing dependency and vulnerability.

#### 3.1.3. Use of the Concept in Nursing Literature

Nåden and Eriksson [13] emphasized that nurses’ understanding of human values and their moral attitudes are essential for the preservation of human dignity, highlighting that moral attitudes, in particular, serve to recognize and safeguard these values. Staats et al. [35] further argued that, in order to implement dignity-preserving care for older female cancer patients with incurable disease residing at home, healthcare professionals must foster a flexible nursing culture, establish effective interprofessional collaboration, and provide individualized care. Ahn [36] emphasized that, for ensuring dignity in the care of terminally ill patients, nurses must prioritize values and moral attitudes, interaction-based communication, the maintenance of comfort, and professional insight and competence.

#### 3.1.4. Dimensions, Attributes, and Measurements in the Theoretical Phase

The literature review and discussions among the research team led to the identification of four common dimensions—namely, the intrinsic, relational, social, and illness-related dimensions—and ten attributes of dignity in care for older cancer patients (Table 2).

Intrinsic dimension

(1)Personal identity

Dignity in care for older cancer patients is grounded in personal identity [37]. Identity represents an inherent, inalienable, unconditional, and absolute human rights.

(2)Awareness of an individual’s intrinsic value

The recognition of an individual’s intrinsic values is crucial for dignity-conserving care. Such recognition encompasses personal values and beliefs that remain intact even in the face of illness or impending death [39].

(3)Deepening sense of value and meaning of life

Essential dignity-conserving care emphasizes the importance of deepening the sense of life’s value and meaning. Older cancer patients, despite the challenges of illness, strive to uphold their quality of life by seeking meaning and purpose, nurturing self-esteem, and anticipating the restoration of health through treatment [18].

(4)Enhancing self-esteem

Dignity-conserving care requires the acceptance of self-esteem. Such acceptance involves strengthening the self, practicing self-care, and restoring self-concept, and it is emphasized as a core element of essential dignity in patients with chronic progressive illness [18]. For older cancer patients, the acceptance of self-esteem is optimized through the enhancement and reinforcement of self-worth.

Relational dimension

(1)Respect

Respect is relational in nature, as it can be either recognized or violated by others. When patients experience respect through courteous and compassionate attitudes, consideration, and acknowledgment of individuality by healthcare professionals, their dignity can be maintained or preserved [38].

(2)Relationships (with medical staff, family, and fellow patients)

Relationships with healthcare professionals, family members, and fellow patients are essential in dignity-conserving care [43]. Older cancer patients preserve or enhance their dignity through the care, comfort, support, encouragement, and reassurance provided by these relationships, thereby highlighting the importance of sharing information by healthcare professionals, emotional bonding with family, and peer support from fellow patients.

Social dimension

(1)Policy support for preserving social dignity

Policy support aimed at preserving dignity in the care of older cancer patients is of critical importance. Respect for the human rights of older adults, along with financial support, facilitates awareness of and access to rights and legal provisions, thereby promoting protective measures that reduce isolation and social marginalization among the elderly [37].

(2)Physical and organizational environmental support within the healthcare system.

Support from societal care and healthcare systems, including physical and organizational environments, is essential for dignity-conserving care. Medical systems providing equipment, facilities, and financial resources constitute the fundamental infrastructure necessary for care [37]. Additionally, the healthcare environment—including staffing levels and the competence of healthcare professionals—is a critical factor in delivering care that promotes health and well-being while preserving and enhancing patient dignity [38].

Illness-related dimension

(1)Free will and choice

Autonomy and the ability to make choices are essential for dignity-conserving care in older cancer patients. Although their dignity may be constrained by illness, when their autonomy in assuming personal roles is respected [15], older patients actively participate in treatment, demonstrating efforts to preserve their roles and pursue independence in the process of recovery.

(2)Appropriate coping strategies

In dignity-conserving care, appropriate coping strategies are crucial for treatment and recovery from illness. As older cancer patients undergo the illness experience and receive medical care, their active participation in treatment and the respect they receive from caregivers enhance adaptive coping strategies, thereby contributing to positive treatment outcomes [43].

Based on these findings, the following tentative definition of dignity in care for older patients with cancer was established during the theoretical phase: *The active pursuit of life value and meaning by older cancer patients, who, despite experiencing dependency and vulnerability due to aging and illness, recognize their identity through respect and care provided by medical staff, family, and fellow patients.*

### 3.2. Fieldwork Phase

The fieldwork phase involved both older cancer patients and the nurses providing their care (Table 2). From the in-depth interviews conducted with the patients, four dimensions and eight attributes of dignity in care were confirmed, with two attributes in each dimension. In the intrinsic dimension, the attributes included personal identity and a deepening sense of value and meaning of life. In the relational dimension, respect and relationships (with medical staff, family, and fellow patients) were identified. The social dimension comprised policy support for caregiving and support from healthcare systems. In the illness-related dimension, free will choice and proactive coping strategies were confirmed.

The FGIs conducted with nurses produced five dimensions and seven attributes. Within the intrinsic dimension, two attributes were highlighted: understanding and respecting human values and ethical and moral attitudes. The relational dimension comprised one attribute: interaction-based communication through relationships. Similarly, the social dimension comprised one attribute: support from healthcare systems. The illness-related dimension included two attributes: protection of dignity and activities promoting dignity. Finally, the professional dimension encompassed one attribute: professional competency.

#### 3.2.1. Older Patients with Cancer (In-Depth Interviews)


**Intrinsic dimension**


(1)Personal identity

Participants reported that while they found it difficult to manage their own bodies, receiving care from others helped them recognize their own worth and develop a positive attitude toward their health. This attitude reflects self-esteem, self-love, and self-worth (i.e., feeling like a valuable person), indicating an awareness of their personal identity. As one participant explained:
*After undergoing this surgery, I felt like I was hit by a bomb. I realized how hard it was to manage my own body. However, through the care I received from others, I came to appreciate the importance of maintaining my health while I was still alive. Even an hour of life feels precious to me now.**(P3)*

(2)A deepening sense of value and meaning of life

Participants reported experiencing higher self-esteem through the care provided by others, which enabled them to recognize the value and meaning of life. This realization motivated them to strive for a healthy and fulfilling life. This attribute highlights the intrinsic and fundamental dignity linked to the pursuit of well-being and the enhancement of life’s value through self-esteem. For example, according to one of the participants:
*When I think about dignity, the thought of how to live and die suddenly hits me. I believe it’s important to find a way to live life independently, no matter what. I take walks on nearby trails to take care of my health. **(P1)*


**Relational dimension**


(1)Respect

Participants experienced enhanced self-esteem and mutual respect through the thoughtful and considerate care provided by the medical staff. For participants, respect involved the recognition of their intrinsic worth in interactions with others, fostering a sense of dignity and enhancing their self-esteem through mutual regard. As one participant noted:
*When receiving care, the nurses preserved my dignity. They did not undress me carelessly and explained procedures thoroughly, allowing me to maintain my confidence as a man and to feel respected. I appreciated their thoughtful care, which ensured I didn’t feel embarrassed. **(P1)*

(2)Relationships (medical staff, family, and fellow patients)

Participants highlighted that during their treatment, the kindness of medical staff, along with the comfort and support provided by family and fellow patients, were among the most valuable and impactful factors. Characterized by the support, encouragement, comfort, and reassurance received from medical staff, family, and fellow patients, these relationships significantly contributed to strengthening their self-esteem and, in turn, reinforcing their sense of dignity. On the subject of relationships, one participant explained:
*The nurses checked on my condition regularly and treated me kindly. My children’s support and encouragement regarding my cancer diagnosis were also very helpful. Such support felt more important than anything else. I feel truly happy that the medical staff, nurses, and my children are always attentive to me. **(P1)*


**Social dimension**


(1)Support for society’s care policies

Participants noted that their physical vulnerability due to illness, compounded by social and economic challenges, underscored the need for robust social care policies. One participant expressed:
*We old people often fall into gaps where we don’t receive proper financial welfare benefits. I hope there will be more balanced support. **(P6)*

This highlights the importance of implementing such policies in a society transitioning toward an aged and super-aged demographic, particularly as such policies are essential for protecting and preserving the dignity of older individuals who are physically and economically vulnerable. It also affirms that social care policies represent an extended support system rooted in the legal and institutional framework designed to safeguard basic human dignity.

(2)Support from healthcare system

Most participants shared that they received financial and personal support from their families during treatment for their illness. However, they stressed the need for additional support from professional medical staff, healthcare facilities, and financial resources, as their spouses, typically of a similar age, faced challenges in providing adequate care. For instance, participants noted that:
*My spouse is too old to provide adequate care for me. I hope hospital-based services can supplement family caregiving. I believe professional medical care would be more effective than relying solely on my spouse. (P4) I believe that the government should provide more financial support, especially for medical expenses. While there are benefits for seriously ill cancer patients, I hope that the financial benefits can be extended to general patients as well. I would like to see more medical services available to the elderly, low-income individuals, and those who are financially disadvantaged. **(P6)*

This underscores that for older individuals who are physically, economically, and environmentally vulnerable due to aging and illness, solely relying on family support is often insufficient. This confirms the critical need for a comprehensive and robust social care system.


**Illness-related dimension**


(1)Free will and choice

Participants expressed satisfaction with their treatment when given the opportunity to make their own decisions regarding their care. They felt respected and experienced a positive treatment journey when empowered to actively participate in decisions about their disease management. Despite their advanced age and associated physical and emotional vulnerabilities, which might have otherwise led them to defer decisions to medical professionals, participants demonstrated a strong sense of self-worth through their autonomy in the decision-making process. As one participant explained:
*The medical staff provided information about costs and treatment options for prostate cancer surgery. I chose open surgery instead of robotic surgery to ensure the complete removal of cancer cells. The medical staff respected my decision and proceeded accordingly. To me, dignity means having the ability to make decisions about my own life. I had a positive experience while receiving surgical treatment for prostate cancer based on my choice. **(P1)*

(2)Proactive coping strategies

Participants reported experiencing pain, anxiety, depression, frustration, and suffering related to their illness. However, they demonstrated the ability to develop effective coping strategies while expressing satisfaction with the care provided. Significantly, older cancer patients highlighted their discomfort with the lack of detailed explanations from medical professionals, emphasizing their strong desire to make informed treatment decisions despite their age. Despite facing significant physical, emotional, and spiritual challenges due to aging and illness, these patients maintained an optimistic outlook regarding their health, bolstered by specific medical information, support, and encouragement from medical staff. As one participant explained:
*When I was diagnosed with prostate cancer, I felt extremely distressed. However, knowing that the cancer had been entirely removed made me happy. It gave me pride and confidence that I could overcome the cancer. It was a game-changer for my attitude toward treatment. **(P1)*
*The information provided by the medical staff guided me in making decisions on several occasions. Although it was a challenging time, as I struggled to understand why I had developed cancer, the support and care from the medical staff helped me maintain a positive outlook. **(P1)*

#### 3.2.2. Nurses (FGIs)


**Intrinsic dimension**


(1)Understanding and respecting human values

Nurses emphasized the importance of prioritizing an understanding of and respect for human values. For instance, one of the nurses asserted:
*Dignity means that everyone, no matter who, is a valuable being who deserves respect. (N4) I believe that everyone has the right to be respected and treated ethically, without any conditions. **(N3)*

This involves acknowledging and valuing each patient’s unique characteristics through a deep understanding of human nature, thereby fostering a foundational attitude of respect for dignity. The humanistic approach and respect demonstrated by clinical nurses help support and strengthen attributes such as self-respect and self-esteem within the intrinsic dimension of older cancer patients’ personal identity.

(2)Ethical and moral attitude

Nurses emphasized the importance of maintaining an ethical and moral attitude through polite behavior, kindness, and consideration, demonstrating continuous interest in and care for their patients. They noted that actively listening to patients and showing kindness helped patients feel respected. For instance, nurses asserted:
*I think we should listen more attentively to patients and show interest in them. It is also important to maintain a kind attitude and explain things clearly and patiently to them. (N4) I believe showing consideration for their well-being is crucial. I once placed drinks and tissues within easy reach for a patient living alone, and she expressed profound gratitude for such a small gesture, saying how thankful she was. **(N3)*

This approach fosters positive self-esteem in older patients with cancer, supporting their pursuit of life’s meaning and purpose. It aligns with the attributes of personal identity and the deepening sense of value and meaning of life identified in the intrinsic dimension of older patients with cancer.


**Relational dimension**


(1)Interaction-based communication through relationships (patient, family, and fellow patients)

Nurses emphasized the need for interaction-based communication in their relationships with the patients, patients’ families, and other patients. They highlighted the need to support patients in the roles they wish to undertake, respect their choices and opinions, and foster close bonds with families and other patients through comfort and encouragement. For instance, nurses asserted:
*I think dignity in care is about respecting patients’ opinions and fostering their independence and autonomy. (N5) When communicating with patients, I believe empathy, respect, and courtesy should guide the conversation. Empathy seems to help patients feel secure. (N10) The role of family seems crucial. When a patient is alone, their ability to manage is limited, but receiving help from family makes them feel gratitude and warmth. **(N7)*

Older patients with cancer develop trust through both verbal and non-verbal communication with nurses, recognizing that they can rely on the nurses when needed. This trust promotes autonomy and independence, facilitating progress toward preserving dignity related to illness. This aligns with the relational dimension attribute of relationships with medical staff for older patients with cancer.


**Social dimension**


(1)Support from healthcare system

Nurses emphasized the need for support from the social healthcare system in caregiving for older adults. One of the nurses asserted:
*The need for caregiving is growing as advancements in medicine extend healthy lifespans in our country. Priority should be given to establishing societal and governmental support systems for the care of older cancer patients.**(N7).*

Another nurse echoed this concern, highlighting systematic gaps in care accessibility:
*I believe there is a significant lack of social and institutional support systems. For example, a public officer mentioned that older cancer patients outside Seoul face challenges in accessing home care services. **(N10)*

Nurses also emphasized the importance of national-level healthcare service support structures in response to the growing demand for aging-related care. This includes both organizational support, such as sufficient staffing and training, and physical support, such as medical equipment, facilities, and financial resources. These needs align with the attributes of policy support for caregiving and social healthcare systems identified in the social dimension of older patients with cancer. The findings underscore the essential role of social policies and healthcare system support in addressing the increasing care needs of older adults with cancer owing to their dependency and vulnerability.


**Illness-related dimension**


(1)Protection of dignity

Nurses reported that they strive to protect patients’ dignity by ensuring physical safety, safeguarding privacy, and maintaining medical confidentiality. Despite their dependency and vulnerability due to illness, patients feel respected when medical staff protect their bodily exposure and respect their personal space through considerate attitudes and actions. As one nurse explained:
*I believe dignity means ensuring that patients do not feel ashamed. For example, when providing medical treatment, drawing a curtain to prevent patients from feeling embarrassed is an act of dignity. **(N1)*

This indicates that older patients with cancer perceive dignity through the respectful behaviors and attitudes of clinical nurses. Such respect empowers patients to actively maintain a sense of self-worth and seek meaning in life, contributing to the restoration of their dignity.

(2)Dignity activities

Nurses summarized their activities as providing medical information, ensuring comfort, and maintaining safety as part of their role as direct caregivers. Clinical nurses reported that older patients feel secure when the care provided supports their physical and emotional stability and reduces suffering through the maintenance of comfort and safety. They also emphasized that providing medical information about treatment empowers older cancer patients to understand their condition better and make informed decisions regarding their care. As one nurse explained:
*I personally explain the disease to older cancer patients. When I share their data and explain test results, they often respond by reflecting on their habits—for instance, saying, “I haven’t been eating well lately, which must have led to these results. I need to eat properly, even when I lack appetite.” This proactive attitude shows that patients understand their illness and take an active role in their treatment decisions. Providing such medical information seems to give them a greater sense of security. **(N9)*

This aligns with the attributes of free will, choice, and proactive coping strategies identified within the illness-related dimension of dignity in care for older cancer patients. As such, nurses’ dignity in care activities involves empowering older cancer patients to actively engage in coping strategies for their treatment, fostering a sense of control and direction. These activities are grounded in positive care experiences provided by medical staff, which help to strengthen the patient’s sense of the value and meaning of life.


**Professional dimension**


(1)Professional competency

Nurses emphasized the importance of professional responsibility, conduct, insight, and competency in delivering dignity in care. Clinical nurses underscored the necessity of understanding the physical, social, and cultural differences among older cancer patients while continuously enhancing their professionalism to provide dignified care. They highlighted the critical role of acquiring up-to-date knowledge, actively participating in relevant training programs, and making deliberate efforts to strengthen their competencies to ensure high-quality care. As one nurse contended:
*I believe that as a nurse, it is my responsibility to build the competencies necessary for delivering dignified care. The quality of dignified care offered to patients, in my opinion, depends significantly on the nurse’s capabilities. Therefore, I feel a strong obligation to develop my skills to provide dignified care for older patients. To achieve this, I believe mandatory or regular ethics education related to dignity is essential. **(N2)*

This perspective reflects the unique role of nurses in preserving and promoting the dignity of older cancer patients, underscoring their responsibility as caregiving professionals committed to safeguarding patient dignity.

### 3.3. Final Analysis Phase

The final analysis phase integrated the findings of the theoretical and fieldwork phases to identify the dimensions, attributes, and indicators of dignified care for older cancer patients. This process involved comparing these attributes with those provided by nurses, culminating in the synthesis of the findings to establish the final definition of dignity in care for older cancer patients. During the course of treatment, older patients with cancer perceived dignity through experiences encompassing the intrinsic, relational, social, and illness-related dimensions. Meanwhile, nurses who provided care for these patients were found to promote patient dignity through care practices that reflected the intrinsic, relational, social, illness-related, and professional dimensions. Based on these findings, a conceptual framework of dignity in care for older patients with cancer was established (Figure 2).

#### 3.3.1. Comparison of Dimensions and Attributes of Dignity in Care for Older Cancer Patients in the Theoretical and Field Work Phases

Most attributes identified in the theoretical phase aligned with those from the fieldwork phase and were consolidated as follows. First, in the intrinsic dignity dimension, the attributes of “personal identity” and “awareness of an individual’s intrinsic value” identified in the theoretical phase were merged into “personal identity,” consistent with the fieldwork phase. Similarly, “deepening sense of value and meaning of life” and enhancing self-esteem” from the theoretical phase were merged into “deepening sense of value and meaning of life,” reflecting the fieldwork findings. Second, in the relational dignity dimension, “respect and relationships (with medical staff, family, and fellow patients)” remained consistent across both phases. Third, in the social dignity dimension, the theoretical phase attributes of “policy support for preserving social dignity” and “physical and organizational environmental support within the healthcare systems” were revised to “support for society’s care policies and support from healthcare systems” in the fieldwork phase. Fourth, in the illness-related dignity dimension, the theoretical phase attribute of “appropriate coping strategies” was renamed “proactive coping strategies” in the fieldwork phase, while “free will and choice” remained unchanged.

For nurses providing dignified care, the intrinsic dignity dimension included the attributes of “understanding and respecting human values” and “ethical and moral attitudes.” In the relational dignity dimension, the attribute identified was “interaction-based communication through relationships.” The social dignity dimension shared the attribute of “support from healthcare systems” with older cancer patients. In the illness-related dignity dimension, the attributes were “protection of dignity” and “engaging in activities promoting dignity.” A new professional dimension comprising the attribute of “professional competency” was identified among nurses.

Antecedents of dignity in care

Empirical indicators are observable measures that clearly represent the attributes of a concept in practical settings. These indicators serve as criteria for assessing the concept of dignity in care for older cancer patients. Based on the attributes identified through individual in-depth interviews with older cancer patients, 17 empirical antecedents for dignity in care were derived (Figure 2).

Additionally, based on the attributes identified through the FGIs with the nurses providing care to these patients, 23 empirical antecedents were derived (Figure 2).

Attributes and consequences of the concept of dignity in care

As a result of this final analysis, the following dimensions and attributes of dignity in care for older cancer patients were identified: personal identity, deepening sense of value and meaning of life, respect, relationships (with medical staff, family, and fellow patients), support for society’s care policies, support from healthcare systems, free will and choice, and proactive coping strategies.

For nurses providing dignified care, seven attributes were identified across five dimensions: understanding and respecting human values and ethical and moral attitudes (intrinsic dimension), interaction-based communication through relationships (relational dimension), support from healthcare systems (social dimension), protection of dignity and engagement in activities promoting dignity (illness-related dimension), and professional competency (professional dimension).

Dignity in care was identified as the preservation, maintenance, and promotion of dignity among older patients with cancer. As they experience illness and receive medical care, when these patients are able to participate actively in their treatment or have their autonomy respected by caregivers, they are more likely to enhance self-management awareness, engage in health-promoting practices, and strengthen their health capacity. Such active coping strategies contribute to improvements in both health outcomes and quality of life.

Defining Dignity in Care for Older Cancer Patients

This study’s final definition of dignity in care for older cancer patients is:
*Although older adults with cancer often experience dependence and vulnerability due to aging and illness, they continue to exercise free will and autonomy, adopting proactive coping strategies and affirming their existential dignity through the respect and compassionate care of healthcare professionals, family members, and peers within the social care system*.

For nurses providing care to older cancer patients, dignity in care is defined as:
*Preserving and enhancing patient dignity by demonstrating professional responsibility and competency as medical professionals delivering direct care through ethical nursing practices based on understanding and respect for human values in interactions and communication with the patients in their charge, families, and other patients*.

#### 3.3.2. Comparison of Dignity in Care in Other Disciplines and the Final Analysis of Dignified Care for Older Cancer Patients

The attributes of dignity in care identified in other disciplines predominantly focus on autonomy and vulnerability within the physical domain. However, the findings of this study reflect the multifaceted experiences of older cancer patients, encompassing physical, intrinsic, external, and social dimensions. By comparing and confirming the attributes of dignity in care based on the understanding and experiences of nurses, this study provides a more concrete and practical explanation. A notable distinction is the identification of professional competence as a unique attribute of nurses providing dignified care. As primary caregivers, clinical nurses proactively preserve and promote the dignity of older cancer patients by demonstrating professional competence.

## 4. Discussion

By comparing and analyzing the experiences and perspectives of older cancer patients and their nurses using a hybrid model of concept analysis, this study elucidated the multidimensional nature and attributes of dignity in care. The attributes of dignified care identified in the theoretical phase were largely consistent with those observed in the fieldwork phase, with certain attributes integrated or refined during the latter. In doing so, this study developed a practical definition of dignity in care for older cancer patients, emphasizing how their sense of free will and adoption of proactive coping strategies can be protected and strengthened by the respect and care provided by social care systems, medical staff, family, and fellow patients. Despite the decline in physical function and dependence on others caused by aging and illness, the older cancer patients interviewed for this study were observed actively pursuing strategies that demonstrate autonomy and competence to achieve true human dignity. Additionally, the nurses delivering care to these patients were found to demonstrate professional responsibility and competency aimed at preserving and enhancing patient dignity through ethical nursing practices based on an understanding of and respect for human values in interactions and communication with these patients, their families, and other patients.

Recognizing their personal identity and striving for the intrinsic dignity of life’s value and meaning, patients experienced improved self-esteem through the kind and considerate attitude of others [48]. Despite suffering from physical symptoms and psychological challenges, such as fear, anxiety, depression, and despair due to cancer [6], as well as limitations on autonomy due to declining physical functions [26], patients reported realizing their worth through the respect shown by others and sought to uphold and value themselves.

During the FGIs, nurses highlighted the importance of understanding human values and showing respect, as well as adopting ethical and moral attitudes [36]. Such ethical respect and moral behavior help preserve dignity by fostering a consolidated understanding of humanity and its inherent value [13]. Practices like empathetic listening, expressing concern, and demonstrating consideration were identified as fundamental principles in nursing [10], contributing to the reinforcement of patients’ self-esteem and a strong sense of dignity.

In this study, older cancer patients expressed the need for support through both social care policies and the healthcare system. They reported relying primarily on their families for both practical and financial assistance during treatment, and often felt guilty and reluctant to burden their spouses and children with the associated financial strain [38]. As their spouses were also older adults, they experienced difficulties in providing or receiving adequate care. Nurses similarly emphasized the need for systemic support—such as funding, equipment, facilities, and personnel—to address the challenges posed by an aging population [37].

These findings align with prior studies highlighting the importance of legal and institutional protections, well-trained staff, and adequate resources in preserving the dignity of older and terminally ill patients [37]. Therefore, societal caregiving policies are integral to safeguarding the dignity of older individuals, especially those vulnerable due to illness and economic hardship.

Older cancer patients interviewed in this study reported being satisfied with their treatment when granted the authority and autonomy to make decisions regarding their cancer treatment [1,11,14,26,36]. This satisfaction with the care they received, coupled with the opportunity to exercise free will and adopt proactive coping strategies, enabled them to actively pursue true dignity during their treatment and recovery process. These findings align with previous research indicating that individuals achieve genuine dignity when they recognize their right to self-realization and are placed in situations where they can utilize their abilities effectively [3].

In the FGIs, nurses, as primary caregivers, reported actively protecting patients’ dignity through actions such as protecting them from bodily exposure, ensuring privacy, providing medical information, and respecting patients’ autonomy in decision-making [11]. Physical protection and privacy are a form of social dignity that safeguards human respect and is preserved in healthcare environments by caregivers [1]. Nurses provided care that upheld patients’ rights to make informed decisions regarding their treatment by offering relevant health information and involving them in the decision-making process. This approach aligns with research findings indicating that supporting patients’ self-determination in choosing and deciding on their treatment helps preserve and enhance their dignity [13]. Corresponding with previous research confirming that dignity is achieved when individuals effectively exercise their abilities [3], despite their vulnerability due to illness or physical disability, older cancer patients who participated in this study actively engaged in treatment-related decision-making, demonstrating autonomy and independence, thereby indicating a strong sense of dignity and self-worth.

In the literature, patients who did not receive direct consultations or detailed explanations about their treatment reported significant discomfort [8]. Patients expressed deep appreciation for the medical staff who provided explanations that alleviated their concerns, enabling them to better understand their condition and treatment options [26]. This underscores the critical importance of verbal and non-verbal communication between medical staff and patients in helping patients obtain medical information [22] and perceive their care as dignified [48]. Effective communication fosters a sense of respect and reassurance, enhancing patients’ perception of dignity. Moreover, as inadequate behaviors, attitudes, or insufficient medical information from caregivers can compromise patient dignity, clinical practice must prioritize involving older patients in treatment decision-making while providing encouragement and support to ease the burden of illness-related concerns.

While experiencing pain, anxiety, depression, and suffering due to cancer [8], older cancer patients adopted proactive coping strategies, nurturing positive hope and aspirations regarding their treatment through empathy, communication, emotional support, and encouragement provided by medical staff. This finding aligns with previous research indicating that dignity can aid cancer patients in preserving the value of life by helping them overcome physical and psychological distress and reduce their will to survive [49]. In this study, patients also expressed feelings of guilt and concern for their families while grappling with the physical and emotional suffering associated with life-threatening cancer [26].

Dignity is perceived or formed as intrinsic dignity through self-esteem and as social dignity through respectful relationships with others [1]. As the most fundamental social relationship built on intimacy and trust through bonds of love, the family was found to play a crucial role, with patients reflecting on their lives and expressing longing or concerns as they faced the separation brought about by death. Additionally, patients reported receiving emotional support and comfort from fellow patients, expressing gratitude for their kindness and care [44]. Older cancer patients, often vulnerable to illness due to aging and underlying chronic diseases, factors that contribute to extended hospital stays [8], preserve their dignity through the respect and empathy they receive from other patients. Research has shown that dignity enhances people’s ability to find meaning in life and overcome illness [17]. The positive bonds formed with fellow patients likely strengthen older cancer patients’ self-esteem and have a beneficial influence on their treatment experiences.

Nurses emphasized the importance of professional responsibility, actions, and competency in delivering dignified care [36], demonstrating their ability to preserve and enhance patient dignity. To provide practical nursing care, nurses support patients using suitable resources tailored to their clinical condition and personal needs [50], engaging in nursing practices that support health recovery, including self-care and social and spiritual support [36]. Through their interactions with caregivers, patients derive a positive self-image and a sense of enduring value, making nurses’ awareness of and commitment to dignified care critical in healthcare settings [51]. In a dignity-related study, Ahn [36] similarly emphasized the significance of nurses’ professional competencies in helping patients feel respected and valued during treatment. Older cancer patients aspire to maintain their quality of life and achieve health recovery through treatment [19]. They actively work to overcome physical and psychological suffering, striving to preserve the meaning and value of life [7], highlighting the critical role of nurses in protecting and enhancing patient dignity.

This study highlights that dignity in care for older cancer patients is sustained through the exercise of free will, personal choice, and active coping strategies, despite the challenges of aging and illness. Such dignity is reinforced by respect and support from healthcare providers, families, and peers, while nurses play a pivotal role through ethical practice and value-based communication. In conclusion, within the context of Korea’s rapidly aging society, dignity in care can be defined as an integrative process encompassing personal, social, cultural, and multidimensional aspects. At its core, this process involves actively engaging older cancer patients in their treatment, providing accessible information about illness, and supporting them through social care policies. Such efforts contribute not only to improved health and quality of life but also to fostering self-management and self-realization.

### 4.1. Limitation

This study has several limitations. As the research was conducted with older cancer patients and nurses in Korea, the participants’ experiences may not fully represent those of all clinical settings, and thus the findings cannot be generalized to all older cancer patients. Therefore, further in-depth and repeated studies that reflect diverse characteristics across countries, regions, and healthcare institutions are needed to better understand perceptions and attitudes toward dignity in care and to promote the well-being and quality of life of older cancer patients. Moreover, by including a broader range of care providers—such as physicians, social workers, and caregivers—future research could identify differences and specific characteristics of dignity in care and thereby provide diverse applications for clinical practice aimed at preserving and enhancing patient dignity.

### 4.2. Conclusion

This study provides an in-depth analysis of the attributes, antecedents, and consequences of dignity in care for older cancer patients. The findings show that, despite the vulnerabilities of aging and illness, dignity is sustained through personal choice, free will, and active coping strategies, which enhance health and quality of life. These insights underscore the significance of dignity in care for healthcare professionals, providers, and policymakers in promoting health and well-being within an aging society. For nurses, dignified care constitutes the foundation of ethical practice and signifies the advancement of professional nursing. Although older patients often experience diminished autonomy, caregiver support enables them to maintain a positive self-image and personal value, thereby strengthening their capacity to cope with illness. Nursing responsibility and competence in providing dignified care require integrating clinical and personal dimensions of care, ensuring access to resources, and supporting recovery. Ultimately, this approach fosters respect for humanity, reinforces ethical and moral practice, and advances professional nursing by enhancing individual nurses’ competencies.

## Figures and Tables

**Figure 1 healthcare-13-02935-f001:**
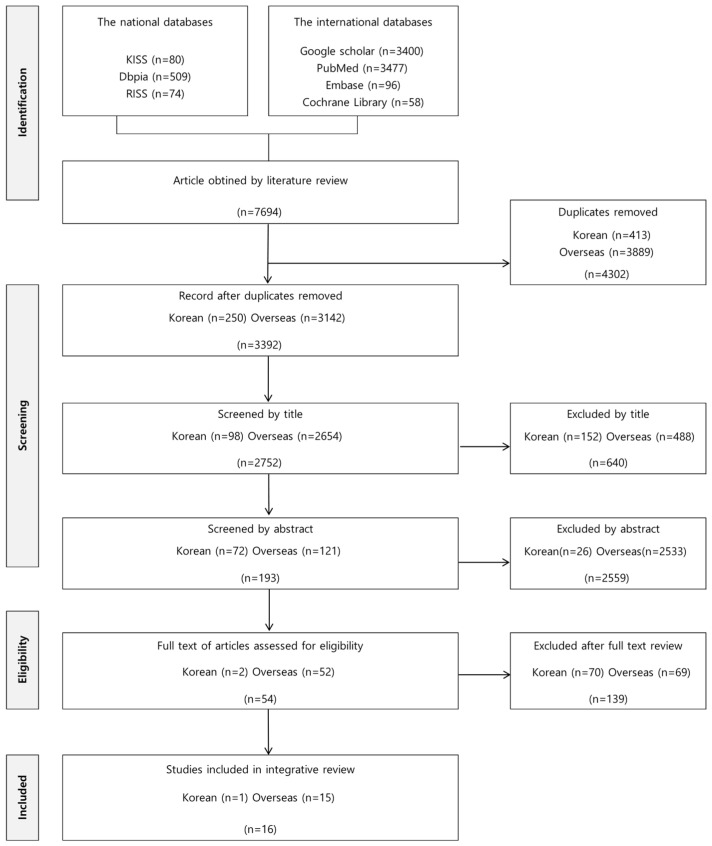
Literature selection process.

**Figure 2 healthcare-13-02935-f002:**
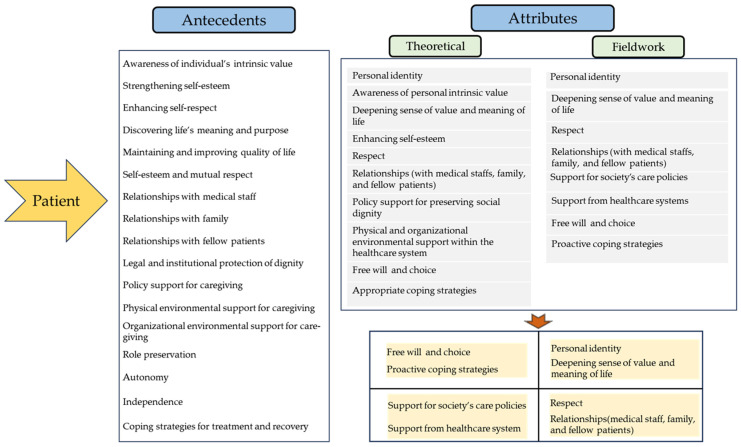
Attributes, antecedents, consequences, and conceptual framework of dignity in care.

**Table 1 healthcare-13-02935-t001:** Participant characteristics. (N = 16).

No	Subjects	Age Group	Departments	Categories		
			(Present)	Diagnosis (Cancer)	Total Nursing Experience (yr)	Nursing Experience in Caring for Older Patients with Cancer (yr)
1	Patients	90s	Ward	Colon		
2	Patients	80s	Ward	Gastric		
3	Patients	70s	OPD	Prostate		
4	Patients	70s	Ward	Ovary		
5	Patients	70s	Ward	Ovary		
6	Patients	70s	Ward	Gastric		
7	Nurse	50~59	OPD		27	5
8	Nurse	40~49	OPD		17	11
9	Nurse	30~39	OPD		14	12
10	Nurse	30~39	OPD		11	7
11	Nurse	30~39	OPD		9	9
12	Nurse	20~29	Ward		3	3
13	Nurse	20~29	Ward		2	2
14	Nurse	20~29	Ward		2	2
15	Nurse	20~29	Ward		3	3
16	Nurse	20~29	Ward		1	1

**Table 2 healthcare-13-02935-t002:** Dimensions and attributes in theoretical and fieldwork stages.

Dimensions	Theoretical Stage			Fieldwork Stage	
	Attributes	Sources		Attributes	
		Literature	Country	Patient	Nurse
Intrinsic	Personal identity	Banerjee et al. [37]	Canada	Personal identity	Understanding and respecting human values
		Cleland et al. [38]	Australia		
		Pageau et al. [39]	Canada		
		Šaňáková and Čáp [40]	Slovakia		
		Van et al. [41]	Netherlands		
	Awareness of personal intrinsic value	Ahn and Oh [36]	Korea		Ethical and moral attitudes
		Coventry [15]	USA		
		Pageau et al. [39]	Canada		
		Simões and Sapeta [17]	Portugal		
		Van et al. [41]	Netherlands		
	Deepening sense of value and meaning of life	Banerjee et al. [37]	Canada	Deepening sense of value and meaning of life	
		Igai [18]	Japan		
		Xiao et al. [42]	China		
	Enhancing self-esteem	Igai [18]	Japan		
		Xiao et al. [42]	China		
Relational	Respect	Banerjee et al. [37]	Canada	Respect	
Relational	Respect	Cleland et al. [38]	Australia	Respect	
		Xiao et al. [42]	China		
	Relationships (with medical staff, family, and fellow patients)	Allande-Cusso et al. [43]	Spain	Relationships (with medical staff, family, and fellow patients)	Interaction-based communication through relationships (with patient, family, and fellow patients)
		Coventry [15]	USA		
		Igai [18]	Japan		
		Šaňáková and Čáp [40,44]	Slovakia		
		Van et al. [41]	Netherlands		
		Xiao et al. [42]	China		
		Zamanzadeh et al. [45]	Iran		
Social	Policy support for preserving social dignity	Banerjee et al. [37]	Canada	Support for society’s care policies	
	Physical and organizational environmental support within the healthcare system	Banerjee et al. [37]	Canada	Support from healthcare systems	Support from healthcare systems
Social	Physical and organizational environmental support within the healthcare system	Cleland et al. [38]	Australia	Support from healthcare systems	Support from healthcare systems
		Igai [18]	Japan		
Illness-related	Free will and choice	Coventry [15]	USA	Free will and choice	Protection of dignity
		Bagherian et al. [46]	Iran		
		Banerjee et al. [37]	Canada		
		Cleland et al. [38]	Australia		
		Franco et al. [7]	Portugal		
		Šaňáková and Čáp [40,44]	Slovakia		
	Appropriate coping strategies	Allande-Cusso et al. [43]	Spain	Proactive coping strategies	Activities promoting dignity
		Ahn and Oh [36]	Korea		
		Banerjee et al. [37]	Canada		
		Cleland et al. [38]	Australia		
		Igai [18]	Japan		
		Šaňáková and Čáp [40,44]	Slovakia		
Illness-related	Appropriate coping strategies	Simões and Sapeta [17]	Portugal	Proactive coping strategies	Dignity activities
		Taghinezhad et al. [47]	Iran		
		Xiao et al. [42]	China		
Professional					Professional competency

## Data Availability

Data is contained within the article.

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
