# Peer review of "Dignity in Care of Older Patients with Cancer in Korea: A Hybrid Model Concept Analysis"

_healthcare, 2025, doi:10.3390/healthcare13222935_

Round 1
Reviewer 1 Report
Comments and Suggestions for Authors
Thank you for the opportunity to review the manuscript for this ambitious multi-component study. The authors chose the hybrid model of concept analysis to explore dignity in care for older adults with cancer, which included a theoretical phase, involving review of relevant literature identified in a systematic search, and a fieldwork phase, involving analysis of data from interviews with patients and focus groups with nurses. This is a novel approach to gaining insight and understanding of this critical aspect of care for older adults, an increasing demographic within our cancer care systems.
I commend the authors for their thoughtful and rigorous approach to understanding the multiple dimensions and attributes of dignity in care from the perspective of patients and nurses, supported with rich quotes. However, clarification is needed regarding critical concepts, methods, and presentation of findings to support the scientific soundness and overall merit of this work.
Title, Highlights, and Abstract
- I encourage the authors to ensure the labels that they use in the highlights and abstract are consistent with those in the text (e.g., intrinsic instead of personhood).
- In the title the authors refer to “a hybrid model,” suggesting that they developed a hybrid model. However, throughout the text, the authors refer to “the hybrid model” as a method of concept analysis. I suggest they clarify this in the title, perhaps changing to “Dignity in Care of Older Patients with Cancer: A Hybrid Model Concept Analysis”
Introduction
- Line 60: The authors describe older cancer patients as “more vulnerable” than younger patients. More vulnerable to what? More vulnerable to loss of dignity? To loss of autonomy? To mistreatment?
- Line 67-68: I am unclear how nuclear family structures would increase bereavement-related loss. Could this be clarified?
- Throughout the manuscript the authors use the term “dignity in care” to both describe patients’ experience of dignity while receiving care and nurses’ provision of care that enhances patients’ experience of dignity. This distinction is not fully clear until the definitions are presented. I encourage the authors to clarify this distinction in the introduction.
- This may help to clarify the statement in lines 82-83. It is a somewhat tautological statement, that dignity in care is a key factor of dignity in nursing contexts.
- This may also clarify the relationship between the terms “patient dignity” and “dignity in care” as they are used in the first paragraph on p.3.
- Lines 122 – 123: Are the authors overstating the relationship between dignity and treatment outcomes or quality of life? Is there specific research supporting this relationship? If the research is not specific to older adults with cancer, I recommend the authors change this statement to “which may have a significant impact on. . .”
- The study methods are unnecessarily repeated at the end of the introduction and the beginning of the methods in the design section.
Methods
Theoretical Phase
- The search strategy included in the text is very broad, searching for articles that related to older adults OR neoplasms OR dignity, rather than looking for articles that combined these concepts. This search would have pulled any article relating to cancer, which is not consistent with the number of articles that were identified. I encourage the authors to verify if their search strategy is correctly represented in the text.
- What were the inclusion criteria for the selected articles?
Fieldwork Phase
- From where were participants recruited? Table 1 refers to ward and OPD but there no further information about the setting is provided.
- What were the inclusion/exclusion criteria for participants?
- Table 1: I am unclear what “total careers” means. Is this total years as a nurse or as a nurse in cancer care? Also, is the final column related to older adults or older adults with cancer?
- Was data from nurses and patients analyzed separately or together?
- In what language were the interviews/focus groups conducted? How was translation of data managed/verified? For example, in the Korean dictionary definition, the term “inalienably” is used, which is an unusual word. Is this directly from the Korean dictionary, or is it a translation? If it is a translation, who or what program did the translation?
- Line 264: Neutrality is generally not consistent with a qualitative approach, which generally encourages reflexivity about the researcher perspectives that may impact results. I encourage the authors to re-consider neutrality as an element of rigor.
- The term “content validity” is usually associated with the development of questionnaires and scales. I encourage the authors to provide a reference for how this may apply to qualitative data collection.
Results
Theoretical Phase
- The authors state that they included 16 articles in the final review, however, it is difficult to identify these articles. Articles are repeated multiple times in Table 2, so it is difficult to count if this Table includes all 16 articles. Additional references are cited in Section 3.1.4. on the page after Table 2, which are not included in the Table. It is unclear if these are part of the included 16 articles or not. Given that this section is defining the dimensions and attributes I expect that this would be based on articles that were included from the literature review. It would be difficult to justify use of non-included articles to support the definitions.
- Table 2 presents a valuable summary of the dimensions and attributes across phases of the study; however, for English readers, presenting information from left to right is important for clarity. I suggest two possibilities: (1) Simply reverse the columns from left to right, i.e., dimensions in the left-most column, then theoretical stage (attributes then sources), then fieldwork stage. In this case, the definitions from Section 3.1.4. (p. 1 of 29 following Table 2) could be incorporated into the table, decreasing the length of the text, or (2) Keep the literature source in the left-most column, but have a single row for each literature source, listing all the identified dimensions and attributes for that source in the following columns. This would make it easier to identify the included articles.
Fieldwork Phase
- The results for each dimension and attribute are clearly presented and well supported with appropriate quotes.
Final analysis Phase
- Figure 2 includes valuable integrated information; however, the text is too small to read.
- The authors do not provide a description in the text of the conceptual framework. It seems to integrate the dimensions among patients and nurses. It does not seem to add much to the text; however, without a description, I am unable to understand the meaning fully. If it is kept, I encourage the authors to make the conceptual framework a separate, larger figure.
- In this section, the authors return to the term “personhood” rather than “intrinsic” dimension. “Intrinsic” is an adjective, consistent with the other dimension labels. I encourage the authors to be consistent throughout.
- Figure 2 refers to antecedents. Table 3 includes the same items listed as indicators. Antecedents would come before and be required for dignity in care. Indicators would be elements that suggest dignity in care is occurring. These terms are also used interchangeably in section 3.3.1. I encourage the authors to choose one label for these elements and use it consistently throughout.
- Also, the information about antecedents/indicators and attributes is repeated between Figure 2 and Table 3. Both are not needed. I encourage the authors to integrate the information in Figure 2 and Table 3.
- The text is quite lengthy. I encourage the authors to reconsider if the detail in the first two paragraphs of Section 3.3.1 is needed. The organization of these attributes could be made clear in Figure2/Table 3, allowing the authors to reduce the text in this section.
- Some sentences in Section 3.3.1 describe the methods of the final analysis. I encourage the authors to move this information to the methods section and focus Section 3.3.1 on the results.
- In the definition for older cancer patients, the term “recognizing personal identity” is somewhat unclear. Does this refer to perceptions of worth or being worthy?
English
- Overall, the English is quite good, with an occasional unusual word usage (described above).
- I encourage the authors to review the manuscript to reduce repetition and ensure each section is concise.
Reviewer 2 Report
Comments and Suggestions for Authors
Dear authors,
I have reviewed the manuscript entitled "Concept Analysis of Dignity in Care of Older Patients with Cancer: a Hybrid Model."
This study presents a three-phase qualitative approach, beginning with a narrative literature review to identify the attributes and definition of dignity in care. It is followed by a fieldwork phase, in which the theoretical model is validated through practical observations. Finally, the analysis phase synthesizes findings from both previous phases to confirm the conceptual framework.
Title: The title is appropriate and includes the methodology used. The inclusion of the geographical context (Korea) could be considered to enhance specificity, given the potentially significant impact of the sociocultural environment.
- Could the authors consider including geographical context in the title?
Abstract: The abstract is well-written, and summarizes the study's objectives, methods, and key qualitative findings.
Introduction: The introduction provides a solid overview of the concept of dignity from a healthcare perspective. While the background focuses primarily on older patients—given their increased frailty and vulnerability—it is important to emphasize that the concept of dignity is not exclusive to this population. Dignity can be compromised regardless of a patient's age or overall condition, and should be considered a universal concern in clinical care.
- Could the authors reinforce the concept of dignity as a universal human value that may be preserved or violated regardless of the patient's age?
Methods: It provides a detailed description of the three phases of the study.
In my opinion, the results of the literature search (pages 166–170), described in the Methods section, would be more appropriately placed in the Results section as an initial point.
- Could the authors consider relocating the literature search data from the Methods section to the Results section?
Although the authors do not explicitly use the term “triangulation,” their methodological approach aligns with this practice, considering the use of different methods and participant groups (patients and nurses).
- Could the authors consider including a brief discussion on triangulation, data saturation, or internal validation procedures?
Results: The results presentations are clear, including and attributes and dimensions of dignity. I believe that the comparative tables between the theoretical and empirical phases are clear and sufficiently detailed.
Discussion: The discussion section shows an appropriate integration of study findings with existing literature.
Round 2
Reviewer 1 Report
Comments and Suggestions for Authors
Thank you for the opportunity to review this revised manuscript. The authors have clearly worked to address the concerns I raised in my initial review, and significantly clarified their tables and figures.
I do have two remaining concerns.
(1) Search strategy. The authors revised the string presented for international studies. However, as I understand the string presented, their search would have pulled up all articles related to "older patients and neoplasms" or "dignity" or "dignity nursing" or "dignity nursing and? older patients and neoplasms." The search for older patients and neoplasms alone pulls up more than the documented number of articles. In addition, specifying if subject headings or keywords were used, may also clarify this.
(2) The inclusion criteria for the selected articles should be included in the methods. As far as I can see, it was addressed in the cover letter but not included in the manuscript.
Thank you for the opportunity to re-review.
